# Iterative Neural Autoregressive Distribution Estimator (NADE-$k$)

**Tapani Raiko**
Aalto University

**Li Yao**
Université de Montréal

**KyungHyun Cho**
Université de Montréal

**Yoshua Bengio**
Université de Montréal,
CIFAR Senior Fellow

## Abstract

Training of the neural autoregressive density estimator (NADE) can be viewed as doing one step of probabilistic inference on missing values in data. We propose a new model that extends this inference scheme to multiple steps, arguing that it is easier to learn to improve a reconstruction in $k$ steps rather than to learn to reconstruct in a single inference step. The proposed model is an unsupervised building block for deep learning that combines the desirable properties of NADE and multi-prediction training: (1) Its test likelihood can be computed analytically, (2) it is easy to generate independent samples from it, and (3) it uses an inference engine that is a superset of variational inference for Boltzmann machines. The proposed NADE-k is competitive with the state-of-the-art in density estimation on the two datasets tested.

## 1 Introduction

Traditional building blocks for deep learning have some unsatisfactory properties. Boltzmann machines are, for instance, difficult to train due to the intractability of computing the statistics of the model distribution, which leads to the potentially high-variance MCMC estimators during training (if there are many well-separated modes (Bengio *et al.*, 2013)) and the computationally intractable objective function. Autoencoders have a simpler objective function (e.g., denoising reconstruction error (Vincent *et al.*, 2010)), which can be used for model selection but not for the important choice of the corruption function. On the other hand, this paper follows up on the Neural Autoregressive Distribution Estimator (NADE, Larochelle and Murray, 2011), which specializes previous neural auto-regressive density estimators (Bengio and Bengio, 2000) and was recently extended (Uria *et al.*, 2014) to deeper architectures. It is appealing because both the training criterion (just log-likelihood) and its gradient can be computed tractably and used for model selection, and the model can be trained by stochastic gradient descent with backpropagation. However, it has been observed that the performance of NADE has still room for improvement.

The idea of using missing value imputation as a training criterion has appeared in three recent papers. This approach can be seen either as training an energy-based model to impute missing values well (Brakel *et al.*, 2013), as training a generative probabilistic model to maximize a generalized pseudo-log-likelihood (Goodfellow *et al.*, 2013), or as training a denoising autoencoder with a masking corruption function (Uria *et al.*, 2014). Recent work on generative stochastic networks (GSNs), which include denoising auto-encoders as special cases, justifies dependency networks (Heckerman *et al.*, 2000) as well as generalized pseudo-log-likelihood (Goodfellow *et al.*, 2013), but have the disadvantage that sampling from the trained "stochastic fill-in" model requires a Markov chain (repeatedly resampling some subset of the values given the others). In all these cases, learning progresses by back-propagating the imputation (reconstruction) error through inference steps of the model. This allows the model to better cope with a potentially imperfect inference algorithm. This learning-to-cope was introduced recently in 2011 by Stoyanov *et al.* (2011) and Domke (2011).

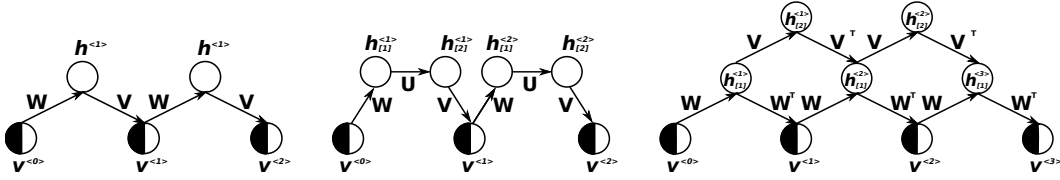

Figure 1: The choice of a structure for NADE-$k$ is very flexible. The dark filled halves indicate that a part of the input is observed and fixed to the observed values during the iterations. Left: Basic structure corresponding to Equations (6–7) with $n = 2$ and $k = 2$. Middle: Depth added as in NADE by Uria *et al.* (2014) with $n = 3$ and $k = 2$. Right: Depth added as in Multi-Prediction Deep Boltzmann Machine by Goodfellow *et al.* (2013) with $n = 2$ and $k = 3$. The first two structures are used in the experiments.

The NADE model involves an ordering over the components of the data vector. The core of the model is the reconstruction of the next component given all the previous ones. In this paper we reinterpret the reconstruction procedure as a single iteration in a variational inference algorithm, and we propose a version where we use $k$ iterations instead, inspired by (Goodfellow *et al.*, 2013; Brakel *et al.*, 2013). We evaluate the proposed model on two datasets and show that it outperforms the original NADE (Larochelle and Murray, 2011) as well as NADE trained with the order-agnostic training algorithm (Uria *et al.*, 2014).

## 2  Proposed Method: NADE-$k$

We propose a probabilistic model called NADE-$k$ for $D$-dimensional binary data vectors $\mathbf{x}$. We start by defining $p_{\boldsymbol{\theta}}$ for imputing missing values using a fully factorial conditional distribution:

$$p_{\boldsymbol{\theta}}(\mathbf{x}_{\text{mis}} \mid \mathbf{x}_{\text{obs}}) = \prod_{i \in \text{mis}} p_{\boldsymbol{\theta}}(x_i \mid \mathbf{x}_{\text{obs}}), \tag{1}$$

where the subscripts mis and obs denote missing and observed components of $\mathbf{x}$. From the conditional distribution $p_{\boldsymbol{\theta}}$ we compute the joint probability distribution over $\mathbf{x}$ given an ordering $o$ (a permutation of the integers from 1 to $D$) by

$$p_{\boldsymbol{\theta}}(\mathbf{x} \mid o) = \prod_{d=1}^{D} p_{\boldsymbol{\theta}}(x_{o_d} \mid \mathbf{x}_{o_{<d}}), \tag{2}$$

where $o_{<d}$ stands for indices $o_1 \ldots o_{d-1}$.

The model is trained to minimize the negative log-likelihood averaged over all possible orderings $o$

$$\mathcal{L}(\boldsymbol{\theta}) = \mathbb{E}_{o \in D!} \left[ \mathbb{E}_{\mathbf{x} \in \text{data}} \left[ - \log p_{\boldsymbol{\theta}}(\mathbf{x} \mid o) \right] \right]. \tag{3}$$

using an unbiased, stochastic estimator of $\mathcal{L}(\boldsymbol{\theta})$

$$\hat{\mathcal{L}}(\boldsymbol{\theta}) = -\frac{D}{D - d + 1} \log p_{\boldsymbol{\theta}}(\mathbf{x}_{o_{\geq d}} \mid \mathbf{x}_{o_{<d}}) \tag{4}$$

by drawing $o$ uniformly from all $D!$ possible orderings and $d$ uniformly from $1 \ldots D$ (Uria *et al.*, 2014). Note that while the model definition in Eq. (2) is sequential in nature, the training criterion (4) involves reconstruction of all the missing values in parallel. In this way, training does not involve picking or following specific orders of indices.

In this paper, we define the conditional model $p_{\boldsymbol{\theta}}(\mathbf{x}_{\text{mis}} \mid \mathbf{x}_{\text{obs}})$ using a deep feedforward neural network with $nk$ layers, where we use $n$ weight matrices $k$ times. This can also be interpreted as running $k$ successive inference steps with an $n$-layer neural network.

The input to the network is

$$\mathbf{v}^{\langle 0 \rangle} = \mathbf{m} \odot \mathbb{E}_{\mathbf{x} \in \text{data}}[\mathbf{x}] + (\mathbf{1} - \mathbf{m}) \odot \mathbf{x} \tag{5}$$

where $\mathbf{m}$ is a binary mask vector indicating missing components with 1, and $\odot$ is an element-wise multiplication. $\mathbb{E}_{\mathbf{x} \in \text{data}}[\mathbf{x}]$ is an empirical mean of the observations. For simplicity, we give

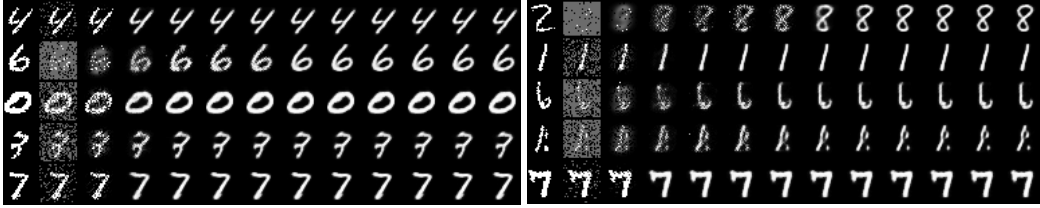

Figure 2: The inner working mechanism of NADE-$k$. The left most column shows the data vectors $\mathbf{x}$, the second column shows their masked version and the subsequent columns show the reconstructions $\mathbf{v}^{\langle 0 \rangle} \dots \mathbf{v}^{\langle 10 \rangle}$ (See Eq. (7)).

equations for a simple structure with $n = 2$. See Fig. 1 (left) for the illustration of this simple structure.

In this case, the activations of the layers at the $t$-th step are

$$\mathbf{h}^{\langle t \rangle} = \phi(\mathbf{W}\mathbf{v}^{\langle t-1 \rangle} + \mathbf{c}) \tag{6}$$

$$\mathbf{v}^{\langle t \rangle} = \mathbf{m} \odot \sigma(\mathbf{V}\mathbf{h}^{\langle t \rangle} + \mathbf{b}) + (\mathbf{1} - \mathbf{m}) \odot \mathbf{x} \tag{7}$$

where $\phi$ is an element-wise nonlinearity, $\sigma$ is a logistic sigmoid function, and the iteration index $t$ runs from 1 to $k$. The conditional probabilities of the variables (see Eq. (1)) are read from the output $\mathbf{v}^{\langle k \rangle}$ as

$$p_{\boldsymbol{\theta}}(x_i = 1 \mid \mathbf{x}_{\text{obs}}) = v_i^{\langle k \rangle}. \tag{8}$$

Fig. 2 shows examples of how $\mathbf{v}^{\langle t \rangle}$ evolves over iterations, with the trained model.

The parameters $\boldsymbol{\theta} = \{\mathbf{W}, \mathbf{V}, \mathbf{c}, \mathbf{b}\}$ can be learned by stochastic gradient descent to minimize $-\mathcal{L}(\boldsymbol{\theta})$ in Eq. (3), or its stochastic approximation $-\hat{\mathcal{L}}(\boldsymbol{\theta})$ in Eq. (4), with the stochastic gradient computed by back-propagation.

Once the parameters $\boldsymbol{\theta}$ are learned, we can define a mixture model by using a uniform probability over a set of orderings $O$. We can compute the probability of a given vector $\mathbf{x}$ as a mixture model

$$p_{\text{mixt}}(\mathbf{x} \mid \boldsymbol{\theta}, O) = \frac{1}{|O|} \sum_{o \in O} p_{\boldsymbol{\theta}}(\mathbf{x} \mid o) \tag{9}$$

with Eq. (2). We can draw independent samples from the mixture by first drawing an ordering $o$ and then sequentially drawing each variable using $x_{o_d} \sim p_{\boldsymbol{\theta}}(x_{o_d} \mid \mathbf{x}_{o_{<d}})$. Furthermore, we can draw samples from the conditional $p(\mathbf{x}_{\text{mis}} \mid \mathbf{x}_{\text{obs}})$ easily by considering only orderings where the observed indices appear before the missing ones.

**Pretraining** It is well known that training deep networks is difficult without pretraining, and in our experiments, we train networks up to $kn = 7 \times 3 = 21$ layers. When pretraining, we train the model to produce good reconstructions $\mathbf{v}^{\langle t \rangle}$ at each step $t = 1 \dots k$. More formally, in the pretraining phase, we replace Equations (4) and (8) by

$$\hat{\mathcal{L}}_{\text{pre}}(\boldsymbol{\theta}) = -\frac{D}{D - d + 1} \frac{1}{k} \sum_{t=1}^{k} \log \prod_{i \in o_{\geq d}} p_{\boldsymbol{\theta}}^{\langle t \rangle}(x_i \mid \mathbf{x}_{o_{<d}}) \tag{10}$$

$$p_{\boldsymbol{\theta}}^{\langle t \rangle}(x_i = 1 \mid \mathbf{x}_{\text{obs}}) = v_i^{\langle t \rangle}. \tag{11}$$

## 2.1 Related Methods and Approaches

**Order-agnostic NADE** The proposed method follows closely the order-agnostic version of NADE (Uria *et al.*, 2014), which may be considered as the special case of NADE-$k$ with $k = 1$. On the other hand, NADE-$k$ can be seen as a deep NADE with some specific weight sharing (matrices $\mathbf{W}$ and $\mathbf{V}$ are reused for different depths) and gating in the activations of some layers (See Equation (7)).

Additionally, Uria *et al.* (2014) found it crucial to give the mask $\mathbf{m}$ as an auxiliary input to the network, and initialized missing values to zero instead of the empirical mean (See Eq. (5)). Due to these differences, we call their approach NADE-mask. One should note that NADE-mask has more parameters due to using the mask as a separate input to the network, whereas NADE-$k$ is roughly $k$ times more expensive to compute.

**Probabilistic Inference** Let us consider the task of missing value imputation in a probabilistic latent variable model. We get the conditional probability of interest by marginalizing out the latent variables from the posterior distribution:

$$p(\mathbf{x}_{\text{mis}} \mid \mathbf{x}_{\text{obs}}) = \int_{\mathbf{h}} p(\mathbf{h}, \mathbf{x}_{\text{mis}} \mid \mathbf{x}_{\text{obs}}) \mathrm{d}\mathbf{h}. \tag{12}$$

Accessing the joint distribution $p(\mathbf{h}, \mathbf{x}_{\text{mis}} \mid \mathbf{x}_{\text{obs}})$ directly is often harder than alternatively updating $\mathbf{h}$ and $\mathbf{x}_{\text{mis}}$ based on the conditional distributions $p(\mathbf{h} \mid \mathbf{x}_{\text{mis}}, \mathbf{x}_{\text{obs}})$ and $p(\mathbf{x}_{\text{mis}} \mid \mathbf{h})$.[1] Variational inference is one of the representative examples that exploit this.

In variational inference, a factorial distribution $q(\mathbf{h}, \mathbf{x}_{\text{mis}}) = q(\mathbf{h})q(\mathbf{x}_{\text{mis}})$ is iteratively fitted to $p(\mathbf{h}, \mathbf{x}_{\text{mis}} \mid \mathbf{x}_{\text{obs}})$ such that the KL-divergence between $q$ and $p$

$$\text{KL}[q(\mathbf{h}, \mathbf{x}_{\text{mis}})||p(\mathbf{h}, \mathbf{x}_{\text{mis}} \mid \mathbf{x}_{\text{obs}})] = -\int_{\mathbf{h}, \mathbf{x}_{\text{mis}}} q(\mathbf{h}, \mathbf{x}_{\text{mis}}) \log\left[\frac{p(\mathbf{h}, \mathbf{x}_{\text{mis}} \mid \mathbf{x}_{\text{obs}})}{q(\mathbf{h}, \mathbf{x}_{\text{mis}})}\right] \mathrm{d}\mathbf{h}\mathrm{d}\mathbf{x}_{\text{mis}} \tag{13}$$

is minimized. The algorithm alternates between updating $q(\mathbf{h})$ and $q(\mathbf{x}_{\text{mis}})$, while considering the other one fixed.

As an example, let us consider a restricted Boltzmann machine (RBM) defined by

$$p(\mathbf{v}, \mathbf{h}) \propto \exp(\mathbf{b}^{\top}\mathbf{v} + \mathbf{c}^{\top}\mathbf{h} + \mathbf{h}^{\top}\mathbf{W}\mathbf{v}). \tag{14}$$

We can fit an approximate posterior distribution parameterized as $q(v_i = 1) = \bar{v}_i$ and $q(h_j = 1) = \bar{h}_j$ to the true posterior distribution by iteratively computing

$$\bar{\mathbf{h}} \leftarrow \sigma(\mathbf{W}\bar{\mathbf{v}} + \mathbf{c}) \tag{15}$$

$$\bar{\mathbf{v}} \leftarrow \mathbf{m} \odot \sigma(\mathbf{W}^{\top}\mathbf{h} + \mathbf{b}) + (\mathbf{1} - \mathbf{m}) \odot \mathbf{v}. \tag{16}$$

We notice the similarity to Eqs. (6)–(7): If we assume $\phi = \sigma$ and $\mathbf{V} = \mathbf{W}^{\top}$, the inference in the NADE-$k$ is equivalent to performing $k$ iterations of variational inference on an RBM for the missing values (Peterson and Anderson, 1987). We can also get variational inference on a deep Boltzmann machine (DBM) using the structure in Fig. 1 (right).

**Multi-Prediction Deep Boltzmann Machine** Goodfellow *et al.* (2013) and Brakel *et al.* (2013) use backpropagation through variational inference steps to train a deep Boltzmann machine. This is very similar to our work, except that they approach the problem from the view of maximizing the generalized pseudo-likelihood (Huang and Ogata, 2002). Also, the deep Boltzmann machine lacks the tractable probabilistic interpretation similar to NADE-$k$ (See Eq. (2)) that would allow to compute a probability or to generate independent samples without resorting to a Markov chain. Also, our approach is somewhat more flexible in the choice of model structures, as can be seen in Fig. 1. For instance, in the proposed NADE-$k$, encoding and decoding weights do not have to be shared and any type of nonlinear activations, other than a logistic sigmoid function, can be used.

**Product and Mixture of Experts** One could ask what would happen if we would define an ensemble likelihood along the line of the training criterion in Eq. (3). That is,

$$-\log p_{\text{prod}}(\mathbf{x} \mid \boldsymbol{\theta}) \propto \mathbb{E}_{o \in D!}\left[-\log p(\mathbf{x} \mid \boldsymbol{\theta}, o)\right]. \tag{17}$$

Maximizing this ensemble likelihood directly will correspond to training a product-of-experts model (Hinton, 2000). However, this requires us to evaluate the intractable normalization constant during training as well as in the inference, making the model not tractable anymore.

On the other hand, we may consider using the log-probability of a sample under the mixture-of-experts model as the training criterion

$$-\log p_{\text{mixt}}(\mathbf{x} \mid \boldsymbol{\theta}) = -\log \mathbb{E}_{o \in D!}\left[p(\mathbf{x} \mid \boldsymbol{\theta}, o)\right]. \tag{18}$$

This criterion resembles clustering, where individual models may specialize in only a fraction of the data. In this case, however, the simple estimator such as in Eq. (4) would not be available.

| Model | Log-Prob. | Model | Log-Prob. |
|---|---|---|---|
| NADE 1HL(fixed order) | -88.86 | RBM (500h, CD-25) | $\approx$ -86.34 |
| NADE 1HL | -99.37 | DBN (500h+2000h) | $\approx$ -84.55 |
| NADE 2HL | -95.33 | DARN (500h) | $\approx$ -84.71 |
| NADE-mask 1HL | -92.17 | DARN (500h, adaNoise) | $\approx$ **-84.13** |
| NADE-mask 2HL | -89.17 | NADE-5 1HL | -90.02 |
| NADE-mask 4HL | -89.60 | NADE-5 2HL | -87.14 |
| EoNADE-mask 1HL(128 Ords) | -87.71 | EoNADE-5 1HL(128 Ords) | -86.23 |
| EoNADE-mask 2HL(128 Ords) | -85.10 | EoNADE-5 2HL(128 Ords) | **-84.68** |

Table 1: Results obtained on MNIST using various models and number of hidden layers (1HL or 2HL). "Ords" is short for "orderings". These are the average log-probabilities of the test set. EoNADE refers to the ensemble probability (See Eq. (9)). From here on, in all figures and tables we use "HL" to denote the number of hidden layers and "h" for the number of hidden units.

## 3  Experiments

We study the proposed model with two datasets: binarized MNIST handwritten digits and Caltech 101 silhouettes.

We train NADE-$k$ with one or two hidden layers ($n = 2$ and $n = 3$, see Fig. 1, left and middle) with a hyperbolic tangent as the activation function $\phi(\cdot)$. We use stochastic gradient descent on the training set with a minibatch size fixed to 100. We use AdaDelta (Zeiler, 2012) to adaptively choose a learning rate for each parameter update on-the-fly. We use the validation set for early-stopping and to select the hyperparameters. With the best model on the validation set, we report the log-probability computed on the test set. We have made our implementation available[2].

### 3.1  MNIST

We closely followed the procedure used by Uria *et al.* (2014), including the split of the dataset into 50,000 training samples, 10,000 validation samples and 10,000 test samples. We used the same version where the data has been binarized by sampling.

We used a fixed width of 500 units per hidden layer. The number of steps $k$ was selected among $\{1, 2, 4, 5, 7\}$. According to our preliminary experiments, we found that no separate regularization was needed when using a single hidden layer, but in case of two hidden layers, we used weight decay with the regularization constant in the interval $\left[e^{-5}, e^{-2}\right]$. Each model was pretrained for 1000 epochs and fine-tuned for 1000 epochs in the case of one hidden layer and 2000 epochs in the case of two.

For both NADE-$k$ with one and two hidden layers, the validation performance was best with $k = 5$. The regularization constant was chosen to be 0.00122 for the two-hidden-layer model.

**Results** We report in Table 1 the mean of the test log-probabilities averaged over randomly selected orderings. We also show the experimental results by others from (Uria *et al.*, 2014; Gregor *et al.*, 2014). We denote the model proposed in (Uria *et al.*, 2014) as a *NADE-mask*.

From Table 1, it is clear that NADE-$k$ outperforms the corresponding NADE-mask both with the individual orderings and ensembles over orderings using both 1 or 2 hidden layers. NADE-$k$ with two hidden layers achieved the generative performance comparable to that of the deep belief network (DBN) with two hidden layers.

Fig. 3 shows training curves for some of the models. We can see that the NADE-1 does not perform as well as NADE-mask. This confirms that in the case of $k = 1$, the auxiliary mask input is indeed useful. Also, we can note that the performance of NADE-5 is still improving at the end of the preallocated 2000 epochs, further suggesting that it may be possible to obtain a better performance simply by training longer.

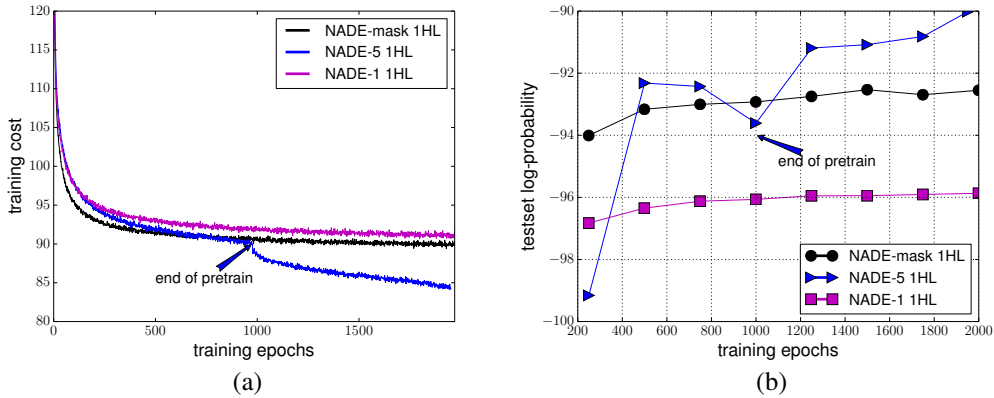

Figure 3: NADE-$k$ with k steps of variational inference helps to reduce the training cost (a) and to generalize better (b). NADE-mask performs better than NADE-1 without masks both in training and test.

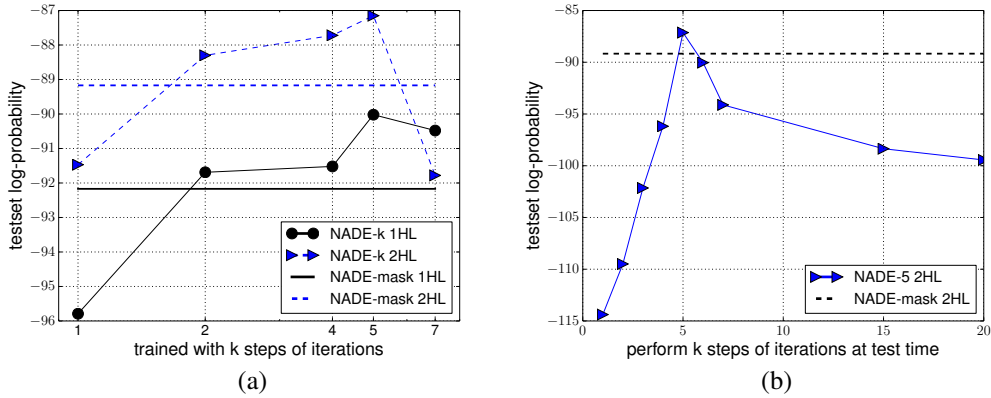

Figure 4: (a) The generalization performance of different NADE-$k$ models trained with different $k$. (b) The generalization performance of NADE-5 2h, trained with k=5, but with various $k$ in test time.

Fig. 4 (a) shows the effect of the number of iterations $k$ during training. Already with $k = 2$, we can see that the NADE-$k$ outperforms its corresponding NADE-mask. The performance increases until $k = 5$. We believe the worse performance of $k = 7$ is due to the well known training difficulty of a deep neural network, considering that NADE-7 with two hidden layers effectively is a deep neural network with 21 layers.

At inference time, we found that it is important to use the exact $k$ that one used to train the model. As can be seen from Fig. 4 (b), the assigned probability increases up to the $k$, but starts decreasing as the number of iterations goes over the $k$. [3]

### 3.1.1 Qualitative Analysis

In Fig. 2, we present how each iteration $t = 1 \ldots k$ improves the corrupted input ($\mathbf{v}^{\langle t \rangle}$ from Eq. (5)). We also investigate what happens with test-time $k$ being larger than the training $k = 5$. We can see that in all cases, the iteration – which is a fixed point update – seems to converge to a point that is in most cases close to the ground-truth sample. Fig. 4 (b) shows however that the generalization performance drops after $k = 5$ when training with $k = 5$. From Fig. 2, we can see that the reconstruction continues to be *sharper* even after $k = 5$, which seems to be the underlying reason for this phenomenon.

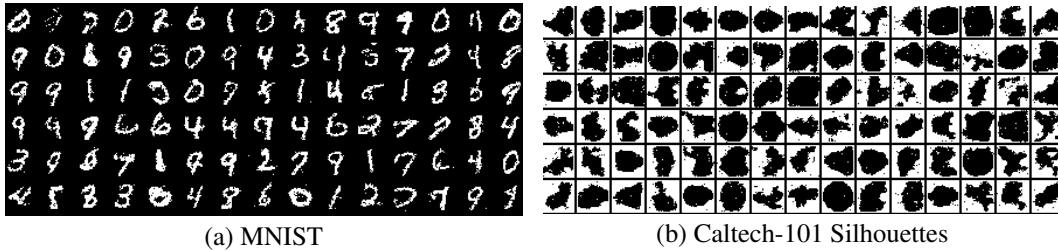

|                    |                         |
|--------------------|-------------------------|
| (a) MNIST          | (b) Caltech-101 Silhouettes |

Figure 5: Samples generated from NADE-$k$ trained on (a) MNIST and (b) Caltech-101 Silhouettes.

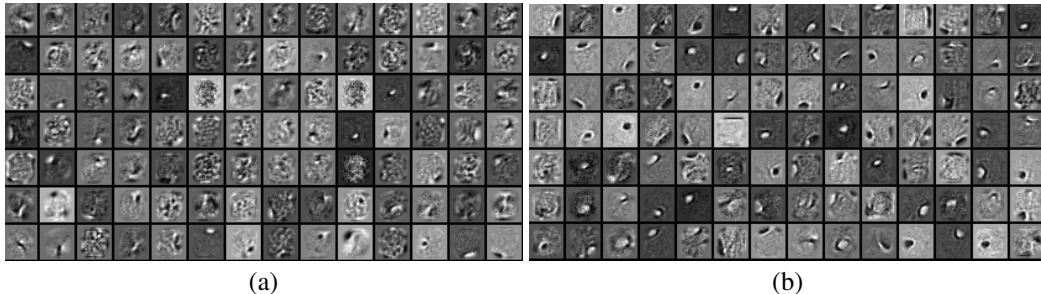

|     |     |
|-----|-----|
| (a) | (b) |

Figure 6: Filters learned from NADE-5 2HL. (a) A random subset of the encodering filters. (b) A random subset of the decoding filters.

From the samples generated from the trained NADE-5 with two hidden layers shown in Fig. 5 (a), we can see that the model is able to generate digits. Furthermore, the filters learned by the model show that it has learned parts of digits such as pen strokes (See Fig. 6).

### 3.1.2 Variability over Orderings

In Section 2, we argued that we can perform any inference task $p(\mathbf{x}_{\text{mis}} \mid \mathbf{x}_{\text{obs}})$ easily and efficiently by restricting the set of orderings $O$ in Eq. (9) to ones where $\mathbf{x}_{\text{obs}}$ is before $\mathbf{x}_{\text{mis}}$. For this to work well, we should investigate how much the different orderings vary.

To measure the variability over orderings, we computed the variance of $\log p(\mathbf{x} \mid o)$ for 128 randomly chosen orderings $o$ with the trained NADE-$k$'s and NADE-mask with a single hidden layer. For comparison, we computed the variance of $\log p(\mathbf{x} \mid o)$ over the 10,000 test samples.

| $\log p(\mathbf{x} \mid o)$ | $\mathbb{E}_{o,\mathbf{x}}\left[\cdot\right]$ | $\sqrt{\mathbb{E}_{\mathbf{x}}\operatorname{Var}_{o}\left[\cdot\right]}$ | $\sqrt{\mathbb{E}_{o}\operatorname{Var}_{\mathbf{x}}\left[\cdot\right]}$ |
|---|---|---|---|
| NADE-mask 1HL | -92.17 | 3.5 | 23.5 |
| NADE-5 1HL | -90.02 | 3.1 | 24.2 |
| NADE-5 2HL | -87.14 | 2.4 | 22.7 |

Table 2: The variance of $\log p(\mathbf{x} \mid o)$ over orderings $o$ and over test samples $\mathbf{x}$.

In Table 2, the variability over the orderings is clearly much smaller than that over the samples. Furthermore, the variability over orderings tends to decrease with the better models.

### 3.2 Caltech-101 silhouettes

We also evaluate the proposed NADE-$k$ on Caltech-101 Silhouettes (Marlin *et al.*, 2010), using the standard split of 4100 training samples, 2264 validation samples and 2307 test samples. We demonstrate the advantage of NADE-$k$ compared with NADE-mask under the constraint that they have a matching number of parameters. In particular, we compare NADE-$k$ with 1000 hidden units with NADE-mask with 670 hiddens. We also compare NADE-$k$ with 4000 hidden units with NADE-mask with 2670 hiddens.

We optimized the hyper-parameter $k \in \{1, 2, \ldots, 10\}$ in the case of NADE-$k$. In both NADE-$k$ and NADE-mask, we experimented without regularizations, with weight decays, or with dropout. Unlike the previous experiments, we did not use the pretraining scheme (See Eq. (10)).

Table 3: Average log-probabilities of test samples of Caltech-101 Silhouettes. ($\star$) The results are from Cho *et al.* (2013). The terms in the parenthesis indicate the number of hidden units, the total number of parameters (M for million), and the L2 regularization coefficient. NADE-mask 670h achieves the best performance without any regularizations.

| Model | Test LL | Model | Test LL |
|---|---|---|---|
| RBM$^\star$ (2000h, 1.57M) | -108.98 | RBM $^\star$ (4000h, 3.14M) | **-107.78** |
| NADE-mask (670h, 1.58M) | -112.51 | NADE-mask (2670h, 6.28M, L2=0.00106) | -110.95 |
| NADE-2 (1000h, 1.57M, L2=0.0054) | -108.81 | NADE-5 (4000h, 6.28M, L2=0.0068) | **-107.28** |

As we can see from Table 3, NADE-$k$ outperforms the NADE-mask regardless of the number of parameters. In addition, NADE-2 with 1000 hidden units matches the performance of an RBM with the same number of parameters. Futhermore, NADE-5 has outperformed the previous best result obtained with the RBMs in (Cho *et al.*, 2013), achieving the state-of-art result on this dataset. We can see from the samples generated by the NADE-$k$ shown in Fig. 5 (b) that the model has learned the data well.

## 4 Conclusions and Discussion

In this paper, we proposed a model called iterative neural autoregressive distribution estimator (NADE-$k$) that extends the conventional neural autoregressive distribution estimator (NADE) and its order-agnostic training procedure. The proposed NADE-$k$ maintains the tractability of the original NADE while we showed that it outperforms the original NADE as well as similar, but intractable generative models such as restricted Boltzmann machines and deep belief networks.

The proposed extension is inspired from the variational inference in probabilistic models such as restricted Boltzmann machines (RBM) and deep Boltzmann machines (DBM). Just like an iterative mean-field approximation in Boltzmann machines, the proposed NADE-$k$ performs multiple iterations through hidden layers and a visible layer to infer the probability of the missing value, unlike the original NADE which performs the inference of a missing value in a single iteration through hidden layers.

Our empirical results show that this approach of multiple iterations improves the performance of a model that has the same number of parameters, compared to performing a single iteration. This suggests that the inference method has significant effect on the efficiency of utilizing the model parameters. Also, we were able to observe that the generative performance of NADE can come close to more sophisticated models such as deep belief networks in our approach.

In the future, more in-depth analysis of the proposed NADE-$k$ is needed. For instance, a relationship between NADE-$k$ and the related models such as the RBM need to be both theoretically and empirically studied. The computational speed of the method could be improved both in training (by using better optimization algorithms. See, e.g., (Pascanu and Bengio, 2014)) and in testing (e.g. by handling the components in chunks rather than fully sequentially). The computational efficiency of sampling for NADE-$k$ can be further improved based on the recent work of Yao *et al.* (2014) where an annealed Markov chain may be used to efficiently generate samples from the trained ensemble. Another promising idea to improve the model performance further is to let the model adjust its own confidence based on $d$. For instance, in the top right corner of Fig. 2, we see a case with lots of missing values values (low $d$), where the model is too confident about the reconstructed digit 8 instead of the correct digit 2.

### Acknowledgements

The authors would like to acknowledge the support of NSERC, Calcul Québec, Compute Canada, the Canada Research Chair and CIFAR, and developers of Theano (Bergstra *et al.*, 2010; Bastien *et al.*, 2012).

## Footnotes

[1] We make a typical assumption that observations are mutually independent given the latent variables.

[2]git@github.com:yaoli/nade_k.git

[3]In the future, one could explore possibilities for helping better converge beyond step $k$, for instance by using costs based on reconstructions at $k - 1$ and $k$ even in the fine-tuning phase.

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
