[Reviews · NeurIPS 2014]

Submitted by Assigned_Reviewer_5

Summary:

The paper introduces an iterative extension of NADE (Neural autoregressive distribution estimator), a generative model that uses a neural network with a variable number of inputs to model each conditional in an autoregressive factorization of a joint distribution. The paper builds up on top of an order-agnostic version of NADE where all dimensions not present in the input are modelled independently by the network at each autoregressive step.

The main idea introduced in the paper is using a prediction of the missing inputs at each iteration, starting with the marginal probability distribution over the training data, and the factorial (each dimension is predicted independently conditioned on the input) approximation obtained from NADE in the following iterations.

The authors hypothesise that prediction in several steps is easier than in one step. A claim they support with experimental results.

NADE-k can be seen as a deep NADE with tied parameters across layers, it would be interesting to see a comparison with NADE-mask that has the same effective number of layers and parameters (example a NADE-mask with 5 layers and h=100 vs NADE-k k=5 h=600).

In the conclusions, the authors suggest adjusting the confidence of intermediate predictions based on d. Shouldn’t NADE already able to do that by outputting values closer to 0.5 when there are few dimensions present in the input? Isn’t it in part the purpose of using masks? For NADE-k, did the authors try using masks at every iteration or just during the first one?

In 3.2 the authors report which regularization methods they tried, but not which worked best, was it the L2 values reported in Table 3?

The complexity of evaluating densities and sampling should be reported. Will the increase in computation be linear with k when using several hidden layers O(DH^2) -> O(kDH^2), but worse when using one hidden layer O(DH) -> O(kDH^2)? This con of the technique should be explicitly reported.

Quality:
The paper is technically sound and the usefulness of the ideas presented is supported by thorough experimental results that obtain state-of-the-art modelling performance on two standard datasets.

Clarity:
The paper is well written.

Originality:
This paper combines ideas (NADE and iterative inference steps) in a novel and pragmatic way. The paper includes a good “related work” section where they ideas they build on are explained and referenced .

Significance:
The paper presents a tractable distribution estimator that obtains state-of-the-art results. The ideas presented will be of interest to the NIPS community, especially for the unsupervised learning community.
Summary: The paper presents an iterative extension of the order-agnostic NADE that obtains significantly better results at a linear increase in computation time. It is an interesting paper with a good explanation of a combination of ideas that work in synergy. The experimental results presented are convincing and show this model obtains state-of-the-art modelling performance.

Submitted by Assigned_Reviewer_33

The paper proposes a simple extension to the inference scheme of the neural autoregressive density estimator (NADE) model.

The key idea is the following: In a NADE model, the conditional probability p(miss | observed) can be viewed as a feed-forward neural network with a single hidden layer, and tied weights going in and out of the hidden layer. The authors propose to define this conditional probability using a multilayer feed-forward neural net with nk layers (n weights that are used k times). Everything else, including using an unbiased stochastic estimator of the negative log-likelihood that uses all possible orderings, is borrowed from Uria et.al.

The paper is well-written. But my main concern is that there is very little novelty in this work.

The authors also conduct experiments on 2 somewhat toyish datasets, MNIST and Caltech-101 silhouettes. The main strength of this paper is that the authors were able to show that on both datasets, their NADE-k model outperforms NADE model that uses 1-step of inference.

Summary: In general, this is a well-written paper. However, given that the authors propose a rather simple extension to the existing model, I would have liked to see a much stronger experimental results on more realistic datasets.

Submitted by Assigned_Reviewer_42

The paper describes a new variant of (order agnostic, deep) NADE called NADE-k that performs k successive reconstructions/inference steps when imputing the unobserved components of the input vector. To train NADE-k, one must sample a subset of the input variables to impute while observing the remaining input variables. The paper includes experiments showing improved log probs over other versions of NADE on binarized MNIST digit generation and Caltech-101 silhouette generation. There are also experiments exploring different values of k and the variance of the imputations with respect to the choice of variable order.

The NADE-k algorithm presented in the paper seems sensible and is for the most part explained clearly. The experiments included are reasonable ones. The NADE-k algorithm is an incremental improvement of deep, order-agnostic NADE and a relatively straightforward one. However, it is more expensive at both training time and test time (since test k is best set to equal training k). Any algorithm based on unrolling inference steps and backpropagating through them can have a k step version of itself created. The question is really how interesting a multi-step variant is and whether it is worth the computational expense. If CD-1 had been introduced without reference to the infinite steps of maximum likelihood and separately from CD-k, would a paper on extending CD-1 to CD-k have been very novel?

One possible experiment not included in the paper that would have been interesting to see is whether slowly increasing k during training works well. In RBM training, to get very good generative performance quickly, it is helpful to switch from CD-1 to CD-k by stepping up k during training. At the start of training, the model is so inaccurate that CD-1 can make rapid progress, but as training continues, more CD steps can help. Would this idea benefit NADE-k?

Somewhat more practical applications of high quality density models would have been welcome to motivate the paper more strongly, perhaps in compression or denoising. Learning a density over binarized MNIST pixels is not a very compelling task. Since the NADE family of algorithms provide tractable, normalized, high-quality density models, an application where those features are essential would have been more interesting.

The paper was mostly clear. There are a few sentence-level issues, such as line 199 "allow to compute" and footnote 2 "for helping better converge."
I had to slow down a lot while reading when I came upon equation 10, it would be clearer if there was a bit more english text introducing it. In general, the paper is well organized.
Summary: The paper presents an incremental extension to order-agnostic NADE. The experiments convincingly demonstrates its advantages in terms of final log-prob.
Author Feedback
Author rebuttal: Dear reviewers,

We would like to thank you for your thorough and insightful comments. Please, see
below for our answers to your comments.

= About the novelty and contribution (R33&R42) =

We agree that one may argue that our work has very little novelty against (Uria
et al., 2014) but also against (Goodfellow et al., 2013). We acknowledge that we
borrowed what we could from both of them and changed only what we had to,
keeping things as simple as possible. However, we would like to emphasize that
the proposed NADE-k is not a simple ad-hoc combination of two methods, rather,
it is well motivated from the principles (such as variational inference and
denoising autoencoders) we discuss in the paper.

Specifically about CD-1 vs. CD-k raised by the Reviewer 42, we would like to
point out that an idea very close to CD-1 was already described in 1988 (see
Fig. 2 of (Hinton&McClelland, 1988)). However, it was only CD-k that led to the
deep learning revolution.

Hinton, Geoffrey E., and James L. McClelland. "Learning representations by
recirculation." NIPS, 1988.

= R33 =

"The authors propose to define this conditional probability using a multilayer
feed-forward neural net with nk layers. Everything else .. is borrowed from
Uria et.al."

- Please, see our answer above.

= R42 =

"If CD-1 had been introduced without reference to the infinite steps of maximum
likelihood and separately from CD-k, would a paper on extending CD-1 to CD-k
have been very novel?"

- Please, see our answer at the beginning of this response letter.

"whether slowly increasing k during training works well."

- Thanks for your suggestion. We agree that slowly increasing k during training
should work well. Our pretraining scheme Eq. (10-11) seemed to work well
enough, too, and it was easier to implement in practice for technical reasons
(not having to change the model structure during training).

"There are a few sentence-level issues"

- Thanks for pointing them out. We will fix them in the final version.

"equation 10"

- We will describe it more clearly in the final version.

= R5 =

"NADE-k can be seen as a deep NADE with tied parameters across layers, it
would be interesting to see a comparison with NADE-mask that has the same
effective number of layers and parameters (example a NADE-mask with 5 layers
and h=100 vs NADE-k k=5 h=600)."

- Uria et al. (2014) reported that using 3 or 4 hidden layers was worse than
using only 2 hidden layers. We do not expect the NADE-mask 5 hidden layers to
work well.

"the authors suggest adjusting the confidence of intermediate predictions based
on d"

- Yes, the mask input could do that. We tried giving the mask only to the first
step (out of k) but it made the learning progress slower. Flipping the
mask-representation bits (1 for missing or observed) changed the performance,
which indirectly suggests that it was probably an optimization issue. We think
not using the mask as an input makes the model more elegant, requires fewer
parameters, and makes the whole inference iteration of NADE-k closer to, e.g.,
variational mean-field fixed-point iteration.

"was it the L2 values reported in Table 3?"

- Yes, the best regularizations are the one reported in Table 3. We apologize
for not mentioning that in the text, and we will make it clearer in the final
version.

"The complexity of evaluating densities and sampling should be reported."

- The complexities should be O(D^2 H)->O(k D^2 H) for one hidden layer and O(D^2
H + D H^2) -> O(k D^2 H + k D H^2) for two hidden layers. There is nothing
specifically worse for the one hidden layer case.